# Role of L-Carnitine supplementation on rate of weight gain and biomarkers of Environmental Enteric Dysfunction in children with severe acute malnutrition: A protocol for a double-blinded randomized controlled trial

Jinat Alam[1], Md. Ridwan Islam[1], Shah Mohammad Fahim [1]*, Md. Amran Gazi[1], Tahmeed Ahmed[1,2,3,4]

1 Nutrition and Clinical Services Division, International Centre for Diarrhoeal Disease Research, Bangladesh (icddr,b), Dhaka, Bangladesh, 2 Office of the Executive Director, International Centre for Diarrhoeal Disease Research, Bangladesh (icddr,b), Dhaka, Bangladesh, 3 Department of Global Health, University of Washington, Seattle, WA, United States of America, 4 Department of Public Health Nutrition, James P Grant School of Public Health, BRAC University, Dhaka, Bangladesh

* mohammad.fahim@icddrb.org

## Abstract

### Background

Severe acute malnutrition (SAM) and environmental enteric dysfunction (EED) are highly prevalent among children residing in resource-limited countries like Bangladesh. L-carnitine may play a role in improving the growth and ameliorating the EED among nutritionally vulnerable children.

### Objective

To investigate the role of L-carnitine supplementation on the rate of weight gain, duration of hospital stays, and EED biomarkers among children with severe acute malnutrition.

### Methods

This study is a double-blinded, placebo-controlled, randomized clinical trial aiming to enroll diarrheal children with SAM between 9–24 months of both sexes attending the nutritional rehabilitation unit (NRU) of Dhaka Hospital of icddr,b. It is an ongoing trial including two arms where one arm receives L-carnitine supplementation, and the other arms receive a placebo for 15 days in addition to the existing standard treatment of SAM. The primary outcome is the rate of weight gain, and the secondary outcomes include duration of hospital stay and EED biomarkers. Outcomes are assessed at baseline and 15 days of post-intervention. We hypothesize that the L- carnitine supplementation for 15 days in children with SAM will improve the rate of weight gain and biomarkers of EED.

**Data Availability Statement:** No datasets were generated or analyzed during the current study. All relevant data from this study will be made available upon study completion.

**Funding:** #Recipient: Jinat Alam #Award Number: BC#21125966 #Full name of Funder: 'Rainy Day Grant Fund' under 'Young investigator's award', primarily financed by the 'International Centre for Diarrhoeal Disease Research, Bangladesh (icddr,b) #No. The funders had and will not have a role in study design, data collection and analysis, decision to publish, or preparation of the manuscript.

**Competing interests:** The authors have declared that no competing interests exist.

## Trial registration

ClinicalTrials.gov # NCT05083637. Date of registration: October 19, 2021.

## Introduction

Globally, severe acute malnutrition (SAM) rates remain alarming, and approximately 14 million children aged less than five years are severely malnourished [1]. Two-thirds of the children burdened with SAM live in Asia, including Bangladesh [1]. As per the latest national survey of Bangladesh (BDHS 2017–18), the burden of childhood malnutrition is still substantial in Bangladesh [2]. Despite having standardized management protocols, half of the total deaths of under-5 children can be attributed to severe malnutrition [1, 3, 4]. Following discharge, it is also associated with high relapse rates [5]. The management of SAM occupies a unique position between clinical medicine and public health, and its management requires special attention [6].

Moreover, children with SAM suffer from essential micronutrient deficiencies. Several research findings suggest that children with SAM possess a decreased serum carnitine level, an essential micronutrient related to nutrition [7]. L-carnitine is a standard biologically active form of carnitine, plays a critical role in the β-oxidation of fatty acids and energy production in the form of adenosine triphosphate (ATP) [7–9]. Carnitine is essential for improving heart and brain function, muscle movement, and many other biological processes within the body [9]. The deficiencies of carnitine may afflict the growth and development of a child of growing age. Some research projects demonstrated that an increased level of serum carnitine has a role in weight gain among these vulnerable populations [7]. L-carnitine may have a potential role in developing and progressing a sub-clinical intestinal disorder termed Environmental Enteric Dysfunction (EED), which is pervasively common in children living in tropical countries [10]. Nutritional impairment in children with SAM can be perilous when carnitine deficiency is present, in addition to EED [11]. EED is characterized by small intestinal inflammation and abnormal gut permeability due to generalized disturbances of small intestinal structure and function. Recent studies suggest that EED in children is associated with a secondary deficiency of carnitine [11]. Also, carnitine deficiency leading to EED may negatively affect young children's growth and cognitive development. However, evidence on carnitine status and its consequences concerning EED in children with SAM and diarrhea is limited in this part of the world [10].

Such a lack of information regarding the role of L-carnitine in improving the rate of weight gain in malnutrition children susceptible to EED is an obstacle in limiting the relapse and adverse consequences of SAM in diarrheal children living in resource-limited countries. Therefore, this study is proposed to assess the carnitine level and explore its role on the rate of weight gain, duration of hospital stays, and EED biomarkers in children with SAM.

### Hypothesis

L- carnitine supplementation for 15 days in children with SAM will improve the rate of weight gain and biomarkers of EED.

### Objectives

The objectives of this study are

1. To investigate the role of L-carnitine supplementation on the rate of weight gain among severely malnourished children;

2. To investigate the role of L-carnitine supplementation on the duration of the hospital stays;

3. To examine the role of L-carnitine supplementation on biomarkers of EED, for instance, myeloperoxidase (MPO), alpha-1 anti-trypsin (AAT), neopterin (NEO), citrulline, and kynurenine: tryptophan (KT) ratio in severely malnourished children.

## Materials and methods

### Trial design

This study is an ongoing double-blinded, placebo-controlled randomized clinical trial, where the children in the intervention arm receive L-carnitine supplementation for 15 days in addition to the existing standard treatment of SAM. And the control arm receives a placebo for the same duration. Both the investigators and study participants and their caregivers don't know who is receiving the particular treatment. In this study, SAM is considered if the Weight for Length Z score (WLZ) is <-3 Standard Deviation (SD) of WHO child growth standards or clinical signs of bilateral pedal edema are present, or the mid-upper arm circumference (MUAC) is <115 mm [12].

### Ethics declaration

Ethical approval was obtained from the Research Review Committee (RRC) and Ethical review committee (ERC) of the International Centre for Diarrheal Disease Research, Bangladesh (icddr,b). (Protocol no: PR-21046; version 1.00; July 29, 2021) (see **S1 File**). Protocol amendments will be given to RRC and ERC if there is any need to modify our protocol. The research protocol follows guidance for protocol reporting, 'the Standard Protocol Items: Recommendations for Interventional Trials (SPIRIT)' (see **S2 File**) [13].

### Study settings and participants

This is a hospital-based intervention study. This study is ongoing in the Dhaka Hospital at icddr,b. This hospital is located at the Mohakhali area in Dhaka city, capital of Bangladesh. This is the largest diarrheal disease hospital in the world. According to recent data, approximately 152,000 patients of all ages are treated here in a year [14]. Among the treated patients, 62% are under the age of 5 years [14]. Usually, patients with diarrheal illnesses and/or associated problems such as electrolyte imbalance, respiratory infection, malnutrition, sepsis, etc., come into this hospital for treatment. There are different wards to treat patients like short-stay unit (SSU) for acute diarrhea, longer stay unit (LSU) for associated morbidity like pneumonia, and NRU for the rehabilitation therapy of malnutrition. Critically ill patients take treatment from this hospital's intensive care unit (ICU), which is well-equipped with necessary life support, including a mechanical ventilator.

In this trial, the study researcher is screening diarrheal children with SAM aged between 9–24 months of both sexes attending LSU. After that, participants are enrolled for this study when they are transferred to the NRU for the rehabilitation phase.

This study includes participants who meet the following eligibility criteria below- (see **Table 1**).

**Table 1. Eligibility criteria.**

| Eligibility criteria |
|---|
| **Inclusion criteria** |
| 1. SAM children with diarrhea aged between 9–24 months |
| 2. Signed informed consent by the parents/caregivers |
| **Exclusion criteria** |
| 1. Septic shock or severe sepsis |
| 2. Participants already taking medications containing L-carnitine |
| 3. Children with Tuberculosis |
| 4. Children with chromosomal anomalies or congenital defects |
| 5. Children with a diagnosed case of Thalassemia |
| 6. Children with an active or previous history of convulsion |

## Sample size and power

The sample size is calculated in this study considering the primary outcome variable. The researcher has considered an interventional study by Alp. Haiden et al. as the primary objective [7]. Using the formula $n = 2 \times \left(\frac{Z1-\alpha+Z1+\beta}{\delta-\delta_{\circ}}\right)^2 \times S^2$, where n is the sample size required in each group, S is the standard deviation of the primary outcome variable = 4.3, $\delta$ is the size of difference of clinical importance = 2.4 (taken from an intervention study where they found that the mean difference between the rate of weight gain in malnourished children after L-carnitine supplementation was 2.4), $\delta_{\circ}$ is the clinically acceptable margin = 0.1, Z1-$\alpha$ is the value for standard normal distribution at 95% level of significance = 1.645 (at a 5% level of significance). Therefore, the final estimated sample size is 49 participants in each group with a 10% attrition rate. And the total sample size for this study is 98. This sample size will allow us to detect significant differences between-arm in our outcomes.

## Randomization

By using a permuted block randomization method with concealment, participants are assigned to this study into two arms (either intervention arm or control arm). This method prevents foreknowledge and makes sure that the random allocation is not made before enrolling a participant to this study. The random allocation sequence was created using a computerized random allocation system for permuted block randomization. It ensures comparable allocation at specific equally spaced points in the sequence of assignments of the participant. A parallel type of randomization was used. Reasonably small blocks with changeable block sizes were built to lower the predictableness. An independent scientist developed this randomization and allocation from icddr,b, who has no connection or participation with this study.

## Blinding

It is a double blinded study. Both the investigators and study participants and their care givers don't know who will be receiving the particular treatment.

## Study recruitment

For enrollment, interventions, and assessments, the researcher of this study follows the Standard Protocol Items: Recommendations for Interventional Trials (SPIRIT) schedule (see **Fig 1**) describes the enrollment schedule, interventions, and assessments, including follow-up throughout the study period. Before enrollment, the field research assistant (study staff) screens every child within the definite age groups according to the eligibility criteria of this trial. Children who fulfill the criteria are brought to the study physician for routine clinical

| Standard Protocol Items: Recommendation for Interventional Trials (SPIRIT) schedule of enrollment, interventions, and assessments | | | | | |
|---|---|---|---|---|---|
| **Protocol Items** | **Study Period** | | | | |
| | Screening (days -3 to -1) | Enrollment (day 0) | Assessment and allocation (day 0) | Intervention (days 0-14) | Follow-up (days 15±3) |
| **Enrollment** | | | | | |
| Eligibility screen | | × | | | |
| Informed Consent | | × | | | |
| Randomization and allocation | | | × | | |
| **Interventions** | | | | | |
| Investigational products | | | | × | |
| Placebo | | | | × | |
| **Assessments** | | | | | |
| Baseline information | × | | | | |
| Physical examinations | × | | | | |
| **Primary Outcome** | | | | | |
| Rate of weight gain | | | × | | × |
| **Secondary Outcomes** | | | | | |
| Duration of hospital stay | | | × | | × |
| EED biomarkers | | | × | | × |

**Fig 1. SPIRIT schedule.**

examination. If participants are eligible for enrollment, the study staff describe the study information in detail, answer any query from the participant's attendant(s), and invite them to enroll their children in the study. If they are interested and willing to give their consent in the study, the children are enrolled after taking written informed consent from their parents or guardians.

## Data collection and management

This study's data collection tools include case report forms, logs, and source documentation. We are collecting relevant information of the participant's such as such as medical history including nature and duration of illness, medication for current illness; socio-demographic characteristics such as age, sex, religion, parental age with education, parent's occupation, monthly family income, number of siblings etc. Information will also be collected about child's feeding practice such as- history of breast feeding, formula or other complementary feeding, immunization status, family history of tuberculosis, recent respiratory tract infection of any family members and past history of child's pneumonia would be recorded coded numbers identify all data, samples, reports, and administrative documents to maintain confidentiality and enable the participant's tracking throughout the period. Labeling with coded numbers is used for all laboratory specimens.

## Consent

In this study, all the enrolled study participant is appropriately treated according to the treatment protocol of this hospital. A signed informed consent statement is collected from each participant's parents, or the authorized legal guardian as the participants are from the minor group. The consenting process occurs before a witness, not the study personnel. No information is remained withheld or concealed from the participants.

## Anthropometric measurements

According to standard operating procedures (SOPs), field research assistants (trained staff) take all the participants' measurements on the day of enrollment and keep all the records in standard CRFs. End-line anthropometry data are taken on the 15th day of supplementation. In addition, on the 180th day after the completion of supplementation, anthropometry measurements are taken to observe the intervention's long-term effect on the rate of weight gain. For measuring weight in kilogram (kg) and length in centimeters (cm), Seca weighing scale and Seca length board are used accordingly for this study. Using a non-stretch tape, the study staff also measures mid-upper arm circumference (MUAC).

## Interventions

In addition to standardized protocolized treatment at the NRU, investigational products and placebo are given to study participants. For any adverse events like diarrhea, nausea, rash, vomiting, or any significant changes in clinical status, study participants are monitored by the physician routinely. Children are treated using appropriate protocolized management at Dhaka Hospital if any adverse events are observed.

After randomization, children of the intervention arm are receiving L-carnitine oral solution (100mg/ml) with 100 mg/kg/day, which is given into 3 divided doses per day for a total of 15 days. On the other hand, children of the control arm receive placebo formulation in the same dose and duration. Placebo solution is identical to the active preparation with no therapeutic value. The investigational products are provided to the study participants at the (NRU) under a controlled set-up.

The flowchart of the overall workflow of this randomized clinical trial is illustrated in below (see **Fig 2**).

## Outcomes measurements

All study outcomes are measured at parallel time points in both arms after completion of interventions (15 ± 3days). Outcome assessments are blinded to the study participant's randomization status.

Primary outcome variable:

i. Rate of weight gain

Secondary outcome variable:
There are two outcome variables, as follows:

i. Duration of hospital stay

ii. EED biomarkers

**Primary outcome measurement.** The primary outcome is the rate of weight gain. The study participant's weight is measured by Seca weighing scale at baseline and after intervention completion. The rate of weight gain (g/kg/day) will be calculated by using the formulae below:

$$\text{Rate of weight gain} = \frac{\text{Weight on completion of study (g)} - \text{Weight on enrolment (or at no edema) (g)}}{\text{Total duration (in days)} * \text{weight on enrolment (kg)}}$$

**Secondary outcomes measurement.** The secondary outcomes include the duration of hospital stays and EED biomarkers.

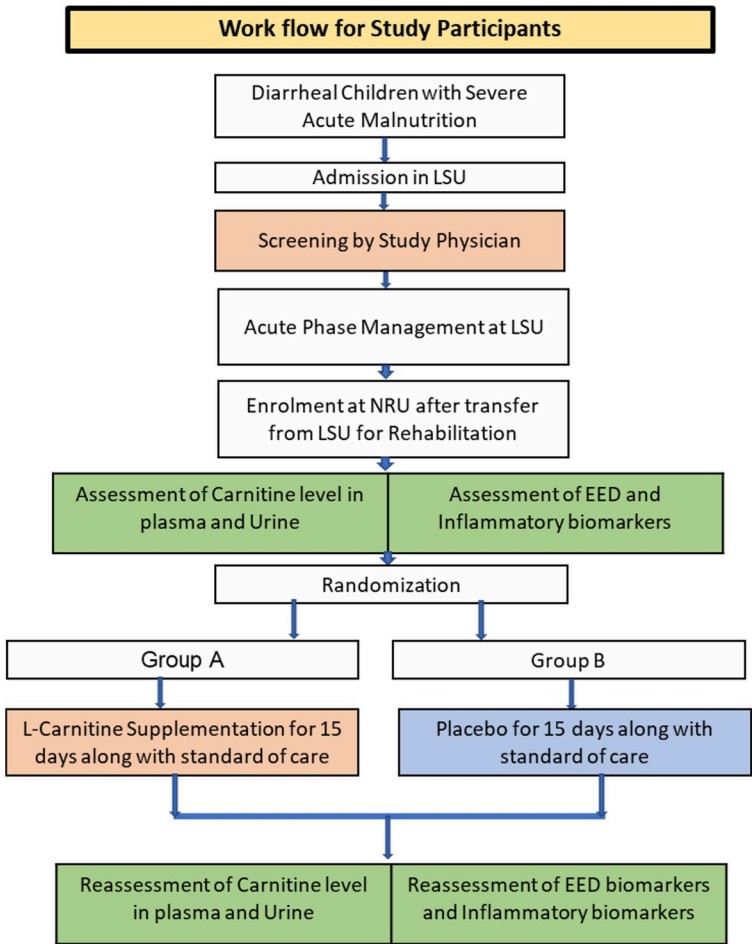

**Fig 2. Workflow for study participants.**

Duration of hospital stay will be assessed by comparing the length of total hospital stay between two arms. Entire inpatient days at NRU are calculated by subtracting the day of admission from the day of discharge from NRU (After exceeding 80% to 85% of the expected value of WLZ and free of edema, children are discharged from NRU) [15].

EED biomarkers will be assessed by doing laboratory investigations of biological samples collected before and after intervention completion.

### Biological sample collection and archiving

All biological samples (blood, urine, feces) are collected, prepared, and preserved as per the SOPs for this protocol. The study staff collects all the biological samples of the study participants on enrollment day and at the end of the nutritional intervention on a follow-up date. Fecal samples are collected and preserved after aliquoted into sterile, prelabelled 2 ml cryovials at the collection site. No additives, preservatives, or media are added to the fecal samples. For blood samples, the study physician collects 5 ml of whole venous blood sample in a blood collection tube aseptically from each participant as per the SOPs. All the samples are immediately placed into a cool box with cool packs. Then the samples are transported to the laboratory. At the laboratory, the blood sample is centrifuged to separate plasma. The plasma is then aliquoted and stored at-80C. In the laboratory, for urine samples preservation, chlorhexidine is

added. During follow-up, all the biological samples are collected from every participant within 3–4 days (window period).

## Laboratory analysis

In this study, investigators are doing all the laboratory investigations (see **Table 2**) at two-time points. On the first day of enrollment, pre-tests are done by the investigators, and the post-test are done after the supplementation of 15 days.

## Data safety monitoring plan

Data safety monitoring is rigorously performed for this study by the monitoring team. After enrollment and data collection, all forms are reviewed again by the study physician, followed by a supervisor for completeness, legibility, and consistency. The supervisor checks randomly to ensure the validity of the collected data by the study physician. According to SOPs, the study physician has trained accordingly for patient screening, enrollment, data collection, and follow-up of study participants.

The study is performed in compliance with the 'Declaration of Helsinki' (2000), the International Council of Harmonization (ICH), Tripartite Guidelines, Guideline for Good Clinical Practice (GCP). These ensure the protection of the study participants' rights and integrity.

## Adverse events

Gastrointestinal symptoms (Nausea, Vomiting, and Diarrhea) are the most likely side effect of the investigational products that are used in this study. This protocol's expected adverse events (AEs) related to the investigational products that do not consider serious adverse events (SAE). SAEs are assessed for severity. All SAEs are being reported within 24 hours of the events to the ERC of icddr,b. Protocol-wise treatment is provided free of cost at Dhaka hospital if medical care is required outside of the protocol.

## Statistical analysis

Data will be presented using frequency with percentages for categorical variables. Mean with standard deviation (SD) will be used for symmetric continuous variables, while median with interquartile range (IQR) will be used for asymmetric numeric variables. The Chi-Square test will be used for comparing the frequency of categorical variables. T-tests or Mann-Whitney U-test will be done to compare the continuous variables between the groups. The Wilcoxon signed-rank test will be used to compare the before and after intervention effects. Data analysis

**Table 2. Laboratory investigations to be performed in the study.**

| Investigations | Biospecimen | Method |
|---|---|---|
| L-carnitine in plasma | Plasma | Enzyme-linked immunosorbent assay (ELISA) |
| Myeloperoxidase (MPO) | Feces | |
| Neopterin (NEO) | Feces | |
| Alpha-1 antitrypsin (AAT) | Feces | |
| Kynurenine: tryptophan (KT) ratio | Plasma | |
| Citrulline | Plasma | |
| C-reactive protein | Plasma | |
| Alpha-1 acid glycoprotein (AGP) | Plasma | |
| Complete blood count (CBC) | Blood | Auto analysis |
| Serum creatinine (S. Cr) | Blood | Enzymatic Photometry |

will be done on the basis of intention-to-treat analysis. All of the participants who would undergo randomization will be included in the analysis. Findings from the withdrawn children will be included in the analysis up to the withdrawal time. A supplementary analysis may also be done, excluding the children withdrawn from this study.

The primary outcome of this trial is the rate of weight gain, while EED biomarkers are the secondary outcomes. We will calculate 95% confidence intervals of the mean change in the rate of weight gain and EED biomarker values between day 0 and day 15. Additionally, multiple linear regression will also be done to determine the factors associated with the rate of weight gain and EED biomarkers in enrolled children. A probability of <0.05 will be considered statistically significant. R software version 4.0.5 will be used to perform the statistical analyses.

## Protocol timeline

The overall study proceedings, starting from staff recruitment and training, patient enrollment and data collection, intervention and laboratory assessment, data entry and analysis with manuscript preparation, are described briefly in below (see **Fig 3**).

## Trial status

Enrollment has been initiated on the 19[th] of 0ctober 2021 and is expected to continue until September 2022 as of April 2022 (see **Fig 3**).

## Discussion and conclusion

The management of SAM occupies a unique position between clinical medicine and public health [16]. This proposal aligns with the 2[nd] and 3[rd] goals of the Sustainable Development Goals (SDGs) [17]. SDG 2 focuses on Zero hunger, and target 2.2 is to end all forms of

| *Study Timeline* (October'2021-September'2022) | | | | | | | | | | | | |
|---|---|---|---|---|---|---|---|---|---|---|---|---|
| **Study Proceedings** | **Oct** | **Nov** | **Dec** | **Jan** | **Feb** | **Mar** | **Apr** | **May** | **June** | **July** | **Aug** | **Sep** |
| Staff recruitment, training, and orientation | | | | | | | | | | | | |
| Patient enrollment and data collection | | | | | | | | | | | | |
| Intervention | | | | | | | | | | | | |
| Laboratory Assessment | | | | | | | | | | | | |
| Data entry, cleaning, and data preparation for analysis | | | | | | | | | | | | |
| Data analysis and interpretations | | | | | | | | | | | | |
| Report writing and preparation of the manuscript | | | | | | | | | | | | |

**Fig 3. Study timeline.**

malnutrition, and SDG 3 focuses on good health and well-being, which perfectly match our research objectives [17]. Moreover, EED and SAM are pervasively common in Bangladeshi children [18]. Recent evidence postulated an association between EED and secondary carnitine deficiency in malnourished children [11]. However, to the best of our knowledge, data on the role of carnitine supplementation in EED and SAM is minimal. Therefore, the results of this study will facilitate us in evaluating the role of L-carnitine supplementation in the rate of weight gain, duration of hospital stays, and EED biomarkers of children who are severely malnourished and received inpatient treatment. Moreover, knowledge of carnitine status and the prevalence of carnitine deficiencies at enrollment in children with SAM will help us design a better treatment strategy for this vulnerable population group. In addition, our study will help to determine the adequate dose and judicial use of L-carnitine oral solution in the facility-based management of SAM.

Various types of issues arose in the development and implementation of this study. All the issues are brought to the investigators' committee for consideration and resolution. Morbidity like pneumonia and poor appetite also represents a significant challenge; counseling resolves these issues. Sometimes familial issues create considerable challenges, that mothers are not complete the rehabilitation phase at NRU. Nowadays, COVID 19 pandemic is also a big challenge for this study [19]. Few of our study participants whose family members were affected by the pandemic couldn't complete the rehabilitation phase, so assessing one of our study outcome-duration of hospital stay would be difficult.

## Supporting information

**S1 File. IRB approved protocol #PR-21046.**
(DOC)

**S2 File. SPIRIT checklist.**
(DOCX)

## Acknowledgments

The authors would like to thank all the participants, their parents, caregivers, and study staff for their valuable contributions. icddr,b also acknowledges the core donors—the governments of Bangladesh, Canada, Sweden, and the UK for providing unrestricted support.

## Author Contributions

**Conceptualization:** Jinat Alam, Shah Mohammad Fahim, Tahmeed Ahmed.

**Funding acquisition:** Jinat Alam.

**Investigation:** Md. Amran Gazi.

**Methodology:** Md. Ridwan Islam, Shah Mohammad Fahim, Md. Amran Gazi.

**Project administration:** Jinat Alam, Shah Mohammad Fahim.

**Resources:** Md. Ridwan Islam, Shah Mohammad Fahim, Md. Amran Gazi, Tahmeed Ahmed.

**Supervision:** Shah Mohammad Fahim, Tahmeed Ahmed.

**Visualization:** Md. Ridwan Islam, Shah Mohammad Fahim.

**Writing – original draft:** Jinat Alam, Shah Mohammad Fahim.

**Writing – review & editing:** Md. Ridwan Islam, Shah Mohammad Fahim, Md. Amran Gazi, Tahmeed Ahmed.

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
