## [Decision Letter · Decision Letter 0]

28 Jun 2022

PONE-D-22-11684Role of L-Carnitine supplementation on rate of weight gain and biomarkers of Environmental Enteric Dysfunction in children with severe acute malnutrition: a protocol for a double-blinded randomized controlled trialPLOS ONE

Dear Dr. Fahim,

Thank you for submitting your manuscript to PLOS ONE. After careful consideration, we feel that it has merit but does not fully meet PLOS ONE’s publication criteria as it currently stands. Therefore, we invite you to submit a revised version of the manuscript that addresses the points raised during the review process.

We look forward to receiving your revised manuscript.

Kind regards,

Walid Kamal Abdelbasset, Ph.D.

Academic Editor

PLOS ONE

Journal Requirements:

4. We note that the original protocol file you uploaded contains a confidentiality notice indicating that the protocol may not be shared publicly or be published. Please note, however, that the PLOS Editorial Policy requires that the original protocol be published alongside your manuscript in the event of acceptance. Please note that should your paper be accepted, all content including the protocol will be published under the Creative Commons Attribution (CC BY) 4.0 license, which means that it will be freely available online, and any third party is permitted to access, download, copy, distribute, and use these materials in any way, even commercially, with proper attribution.

Therefore, we ask that you please seek permission from the study sponsor or body imposing the restriction on sharing this document to publish this protocol under CC BY 4.0 if your work is accepted. We kindly ask that you upload a formal statement signed by an institutional representative clarifying whether you will be able to comply with this policy. Additionally, please upload a clean copy of the protocol with the confidentiality notice (and any copyrighted institutional logos or signatures) removed.

Reviewers' comments:

Reviewer's Responses to Questions

**Comments to the Author**

1. Does the manuscript provide a valid rationale for the proposed study, with clearly identified and justified research questions?

Reviewer #1: Yes

Reviewer #2: Yes

2. Is the protocol technically sound and planned in a manner that will lead to a meaningful outcome and allow testing the stated hypotheses?

Reviewer #1: Yes

Reviewer #2: Yes

3. Is the methodology feasible and described in sufficient detail to allow the work to be replicable?

Reviewer #1: Yes

Reviewer #2: Yes

4. Have the authors described where all data underlying the findings will be made available when the study is complete?

Reviewer #1: Yes

Reviewer #2: Yes

5. Is the manuscript presented in an intelligible fashion and written in standard English?

Reviewer #1: Yes

Reviewer #2: Yes

6. Review Comments to the Author

You may also provide optional suggestions and comments to authors that they might find helpful in planning their study.

Reviewer #1: Dear Author,

Thank you for your very interesting effort. Your finding is very important to the community and I found the detail of your work and it is coined in very excellent manner. Please keep up this fruitful work. Please also check grammatical error in some few place of the work around result and discussion.

Reviewer #2: Review comments on Manuscript Number: PONE-D-22-11684. Entitled " Role of L-Carnitine supplementation on rate of weight gain and biomarkers of Environmental Enteric Dysfunction in children with severe acute malnutrition: a protocol for a double-blinded randomized controlled trial"

Overall, the idea of research is very interesting, well written and reasonable. However, there are some comments and suggestions.

Title : Well structured

Abstract:

- Background is recommended to be more concise.

- Keywords in alphabetical order.

Introduction: Well structured

Methodology : Well structured

Statistical analysis: Well structured

Discussion: Well structured

7. PLOS authors have the option to publish the peer review history of their article (what does this mean?). If published, this will include your full peer review and any attached files.

Reviewer #1: **Yes: **Dr. Habtamu Fekadu Gemede

Reviewer #2: No

---

## [Author Response · Author response to Decision Letter 0]

28 Jul 2022

Response to the Academic Editor's comments:

Response: Dear editor, thank you very much for your comment. File naming was edited to comply with the style requirements. We hopefully have no divergences from the style requirements now.

Response: Dear editor, thank you very much for your suggestion. We moved our ethics statement into the method section after deleting it from the other area (Page:5; Line:100-105).

3. Please include captions for your Supporting Information files at the end of your manuscript, and update any in-text citations to match accordingly. 

Response: Dear editor, thank you for your suggestion. We included captions for our supporting information files at the end of our revised manuscript (Page:18, Line:377-381), and the in-text citation was matched accordingly (Page:5,16; Line:103,105,317).

4. We note that the original protocol file you uploaded contains a confidentiality notice indicating that the protocol may not be shared publicly or be published. Please note, however, that the PLOS Editorial Policy requires that the original protocol be published alongside your manuscript in the event of acceptance. Please note that should your paper be accepted, all content including the protocol will be published under the Creative Commons Attribution (CC BY) 4.0 license, which means that it will be freely available online, and any third party is permitted to access, download, copy, distribute, and use these materials in any way, even commercially, with proper attribution.

Therefore, we ask that you please seek permission from the study sponsor or body imposing the restriction on sharing this document to publish this protocol under CC BY 4.0 if your work is accepted. We kindly ask that you upload a formal statement signed by an institutional representative clarifying whether you will be able to comply with this policy. Additionally, please upload a clean copy of the protocol with the confidentiality notice (and any copyrighted institutional logos or signatures) removed. 

Response: Dear editor, thank you for your advice. We changed the confidentiality notice and agreed to share our protocol publicly.

Response: Dear editor, thank you for your advice. We reviewed our reference list. We hope it is completed and correct now (Page:17-18; Line:333-375).

Response to the Reviewer's comments:

Comments to the Author

1. Does the manuscript provide a valid rationale for the proposed study, with clearly identified and justified research questions?

Reviewer #1: Yes

Reviewer #2: Yes

Response: Dear reviewers, thank you very much for your positive comment.

2. Is the protocol technically sound and planned in a manner that will lead to a meaningful outcome and allow testing the stated hypotheses?

Reviewer #1: Yes

Reviewer #2: Yes

Response: Thank you for your comment.

3. Is the methodology feasible and described in sufficient detail to allow the work to be replicable?

Reviewer #1: Yes

Reviewer #2: Yes

Response: Thank you so much for your valuable opinion.

4. Have the authors described where all data underlying the findings will be made available when the study is complete?

Reviewer #1: Yes

Reviewer #2: Yes

Response: Thank you so much for your comment.

5. Is the manuscript presented in an intelligible fashion and written in standard English?

Reviewer #1: Yes

Reviewer #2: Yes

Response: Thank you very much for your opinion.

6. Review Comments to the Author

Reviewer #1: Dear Author,

Thank you for your very interesting effort. Your finding is very important to the community and I found the detail of your work and it is coined in very excellent manner. Please keep up this fruitful work. Please also check grammatical error in some few place of the work around result and discussion.

Response to Reviewer #1: Thank you for your valuable suggestion and inspiration. We have checked and sorted the grammatical error in the revised manuscript as you suggested (Page:14-15, Line:283-303).

Reviewer #2: Review comments on Manuscript Number: PONE-D-22-11684. Entitled " Role of L-Carnitine supplementation on rate of weight gain and biomarkers of Environmental Enteric Dysfunction in children with severe acute malnutrition: a protocol for a double-blinded randomized controlled trial"

Overall, the idea of research is very interesting, well written and reasonable. However, there are some comments and suggestions.

Title: Well structured

Abstract:

- Background is recommended to be more concise.

- Keywords in alphabetical order.

Introduction: Well structured

Methodology: Well structured

Statistical analysis: Well structured

Discussion: Well structured

Response to Reviewer #2: Thank you for your valuable suggestions. We concised the background of the abstract and reordered keywords alphabetically in the revised manuscript (Page:2-3; Line:26-34, 48-49).

---

## [Decision Letter · Decision Letter 1]

13 Sep 2022

Role of L-Carnitine supplementation on rate of weight gain and biomarkers of Environmental Enteric Dysfunction in children with severe acute malnutrition: a protocol for a double-blinded randomized controlled trial

PONE-D-22-11684R1

Dear Dr. Fahim,

We’re pleased to inform you that your manuscript has been judged scientifically suitable for publication and will be formally accepted for publication once it meets all outstanding technical requirements.

Kind regards,

Walid Kamal Abdelbasset, Ph.D.

Academic Editor

PLOS ONE

Additional Editor Comments (optional):

Reviewers' comments:

Reviewer's Responses to Questions

**Comments to the Author**

1. Does the manuscript provide a valid rationale for the proposed study, with clearly identified and justified research questions?

Reviewer #2: Yes

2. Is the protocol technically sound and planned in a manner that will lead to a meaningful outcome and allow testing the stated hypotheses?

Reviewer #2: Yes

3. Is the methodology feasible and described in sufficient detail to allow the work to be replicable?

Reviewer #2: Yes

4. Have the authors described where all data underlying the findings will be made available when the study is complete?

Reviewer #2: Yes

5. Is the manuscript presented in an intelligible fashion and written in standard English?

Reviewer #2: Yes

6. Review Comments to the Author

You may also provide optional suggestions and comments to authors that they might find helpful in planning their study.

Reviewer #2: Review comments on Manuscript Number: PONE-D-22-11684R1. Entitled "Role of L-Carnitine supplementation on rate of weight gain and biomarkers of Environmental Enteric Dysfunction in children with severe acute malnutrition: a protocol for a double-blinded randomized controlled trial"

I would like to thank the authors for their successful work to address the reviewers' comments. The authors have done great efforts to accomplish this work. They fulfilled all comments and made necessary changes throughput the manuscript. I recommend accepting the manuscript its revised form.

7. PLOS authors have the option to publish the peer review history of their article (what does this mean?). If published, this will include your full peer review and any attached files.

Reviewer #2: No

---

## [Editor Report · Acceptance letter]

22 Sep 2022

PONE-D-22-11684R1 

Role of L-Carnitine supplementation on rate of weight gain and biomarkers of Environmental Enteric Dysfunction in children with severe acute malnutrition: a protocol for a double-blinded randomized controlled trial 

Dear Dr. Fahim:

I'm pleased to inform you that your manuscript has been deemed suitable for publication in PLOS ONE. Congratulations! Your manuscript is now with our production department. 

Kind regards, 

on behalf of

Dr. Walid Kamal Abdelbasset 

Academic Editor

PLOS ONE